# TimeNeRF: Building Generalizable Neural Radiance Fields across Time from Few-Shot Input Views

## ABSTRACT

We present TimeNeRF, a generalizable neural rendering approach for rendering novel views at arbitrary viewpoints and at arbitrary times, even with few input views. For real-world applications, it is expensive to collect multiple views and inefficient to re-optimize for unseen scenes. Moreover, as the digital realm, particularly the metaverse, strives for increasingly immersive experiences, the ability to model 3D environments that naturally transition between day and night becomes paramount. While current techniques based on Neural Radiance Fields (NeRF) have shown remarkable proficiency in synthesizing novel views, the exploration of NeRF's potential for temporal 3D scene modeling remains limited, with no dedicated datasets available for this purpose. To this end, our approach harnesses the strengths of multi-view stereo, neural radiance fields, and disentanglement strategies across diverse datasets. This equips our model with the capability for generalizability in a few-shot setting, allows us to construct an implicit content radiance field for scene representation, and further enables the building of neural radiance fields at any arbitrary time. Finally, we synthesize novel views of that time via volume rendering. Experiments show that TimeNeRF can render novel views in a few-shot setting without per-scene optimization. Most notably, it excels in creating realistic novel views that transition smoothly across different times, adeptly capturing intricate natural scene changes from dawn to dusk.

## CCS CONCEPTS

• **Computing methodologies** → **3D imaging**; *Reconstruction*.

## KEYWORDS

Neural Radiance Field from Sparse Inputs; Volume Rendering; Time Translation

## 1 INTRODUCTION

Novel view synthesis (NVS), an essential challenge in computer vision, aims to synthesize unseen viewpoints from posed images. Its applications range from virtual reality (VR) and augmented reality (AR) to 3D scene reconstruction. Also, it's a crucial technique to achieve the metaverse. The rise of neural rendering techniques, especially Neural Radiance Fields (NeRF) [35] and its successors [29, 36, 50, 64], has ushered in impressive progress in novel view synthesis. However, a notable drawback of these prior works is

*ACM MM, 2024, Melbourne, Australia*
© 2024 Copyright held by the owner/author(s). Publication rights licensed to ACM.
ACM ISBN 978-x-xxxx-xxxx-x/YY/MM
https://doi.org/10.1145/nnnnnnn.nnnnnnn

their reliance on per-scene optimization and the need for hundreds of different viewpoint images. In practical scenarios, input views are often limited, and re-optimizing the model for new scenes is inefficient. Moreover, to achieve complete immersion in virtual reality or the metaverse, creating environments that can transition seamlessly from day to night is essential. Nevertheless, the exploration of NeRF's potential for temporal 3D scene modeling remains limited, with few dedicated datasets available for this purpose.

In this paper, we present TimeNeRF, a framework designed to achieve novel view synthesis in three aspects: 1) a few-shot setting, 2) generalizing to new scenes captured under varying conditions, and 3) handling temporal transitions in 3D scenes. Several studies are devoted to realizing novel view synthesis from sparse viewpoints [20, 40, 49, 58] and also ensuring generalizability to previously unseen scenes [2, 19, 27, 30, 54, 63]. However, when it comes to rendering novel views at different times, the viable option for these methods is capturing an additional set of views at the desired time. On the other hand, NeRF-W [33] and Ha-NeRF [3] explore constructing neural radiance fields from images taken at various times and under different illuminations. Though capable of generating novel views at different times, they still depend on appearance codes from reference images and require many viewpoints and per-scene optimization. Recently, CoMoGAN [43] introduces a continuous image translation model capable of altering an image's appearance to correspond to a different time. Yet, when integrated directly with a novel view synthesis model, it succumbs to the view inconsistency problem, a limitation highlighted in earlier works [6, 15, 16].

Consequently, a unified 3D representation model that generalizes across different 3D scenes over time, especially in a few-shot setting, remains a challenging open question. To address this issue, we introduce TimeNeRF. Distinct from prior approaches, TimeNeRF synthesizes novel views from limited viewpoints without necessitating model retraining for previously unseen scenes. Moreover, by constructing novel time-dependent neural radiance fields, our method can render novel views at specific times without relying on reference images.

The key idea of TimeNeRF, as shown in Fig. 1, is to first construct a content radiance field, then transform it into the neural radiance field of a specific moment by infusing the environmental change information relevant to the desired time into the content radiance field. Specifically, our approach involves a two-stage training process. The first stage focuses on disentangling content and environmental change factors, achieved through an image translation model. The second stage begins by extracting geometry information from a few input views via the appearance-agnostic geometry extractor. We build our model upon previous generalizable NeRF-based methods but introduce a significant adjustment: the cost volume is constructed from content features instead of being extracted from standard convolutional network features, allowing our model to

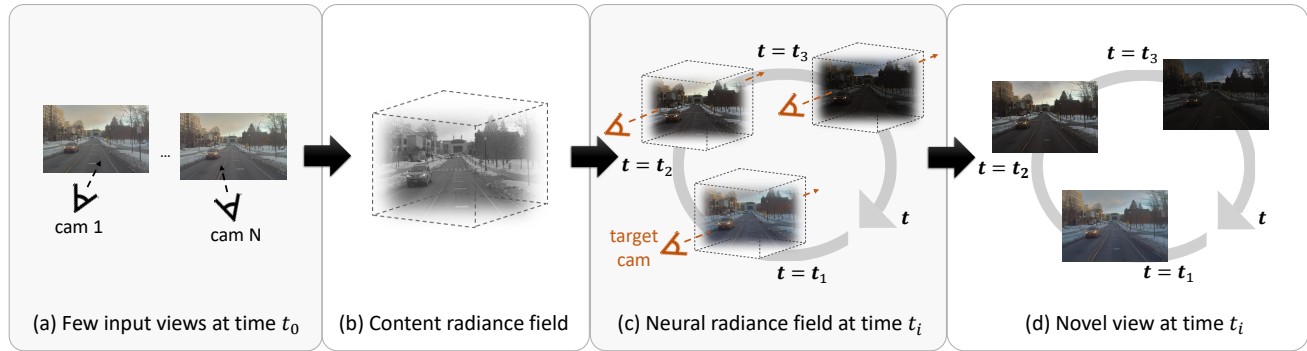

**Figure 1: Overview of TimeNeRF. By inputting few input views (a), our method first constructs a content radiance field that filters out environmental changing factors (b), enabling us to obtain neural radiance fields at arbitrary times (c). Finally, our model seamlessly renders novel views with a smooth transition of time (d), providing an immersive and realistic experience.**

handle various capture conditions. Next, the implicit scene network constructs a content radiance field based on the extracted geometry and features. We then predict the time and extract time-irrelevant features from the style feature obtained from a pre-trained style extractor. Finally, we create time-dependent radiance fields by incorporating the time and the extracted features into the content radiance field for novel view synthesis over time.

To train the model, it is ideal to have a dataset with various capture views and different capture times. However, collecting dense inputs of a scene is expensive, especially considering our specific problem setup, which requires generating novel views at particular times. This means that, in addition to the substantial effort required to gather multiple views of the scene, we also need to collect data at different times throughout the day, which in practice becomes very costly. To overcome this, we train our model on both the Ithaca365 dataset [9], which offers few different capture views but has limited time information, and the Waymo dataset [51], covering a range of capturing times (i.e., day, dusk, dawn, night). Since these datasets do not contain time labels, we train the model without relying on exact time data. Specifically, we leverage our two-stage training approach with the design of our network to map reference images into the time domain. The design also benefits the testing phase, allowing us to render novel views at any time by specifying the time code directly. Additionally, we introduce several loss functions to ensure that our model exhibits cyclic changes and that the transition results are adapted based on the inputs. We will explain the details in the "Proposed Method" section. In general, our main contributions are:

- An extended novel setting for view synthesis over time, which is more practical for real-world applications.
- A novel time-dependent and NeRF-based approach that renders novel views at arbitrary times from few inputs and achieves generalizability to new scenes captured under any conditions.
- Extensive experiments showing TimeNeRF's capability to transition smoothly across different times of the day.

## 2 RELATED WORKS

### 2.1 Neural Radiance Field

Neural Radiance Field (NeRF) [35] revolutionized novel view synthesis by modeling 3D scenes as an implicit neural representation. It employs a multi-layer perceptron (MLP) to map 3D positions and camera directions to colors and densities. Subsequent works improve NeRF by combining implicit scene representation with traditional grid representation [7, 14, 29, 36, 50, 56] or reducing samples taken along each ray [24, 28, 38, 42]. Nevertheless, these techniques are limited in generating views for different time points and face challenges with varied illumination.

NeRF-W [33] extends the NeRF model by introducing a learned appearance code for each image, allowing it to synthesize novel views at different times by inputting different appearance codes. The subsequent works [3, 23, 48, 52] adopt a similar idea. However, these methods require per-scene optimization and a large number of training views. Moreover, they lack the ability to query the view at a specific time.

On the other hand, NeRF to dynamic scene [13, 31, 41, 44, 48] aims to model a dynamic scene by learning a 4D implicit scene representation. The inputs of the MLP include not only 3D coordinates and direction but also time. While these approaches target short-range temporal changes, like moving people or objects, we emphasize modeling long-range time shifts, enabling realistic view rendering from day to night. Furthermore, these methods also require many image views and re-optimizing the model for each new scene.

### 2.2 Few-Shot NeRF

Recently, three major approaches have been proposed to enable the synthesis of novel views from limited inputs. First, some methods [17, 20, 40, 49, 58, 61] incorporate semantic and geometry regularizations to constrain the output color and density. Second, another line of works attempts to leverage additional depth information such as sparse 3D points [8, 46] or depth prediction from a pre-trained model [12]. Third, some other approaches [2, 19, 27, 30, 39, 54, 57, 60, 63] attempt to condition the model with features extracted from inputs, allowing it to be generalizable to new scenes. For example,

MVSNeRF [2] proposes incorporating a 3D cost volume constructed from the extracted feature of input views [11, 62] with the NeRF model for assistance. This allows the model to develop geometry awareness of scenes and increases model generalization ability. Later works [19, 30, 39, 60] also adopt this idea. GeoNeRF [19] and NeuRay [30] further take the occlusion problem into account by predicting the visibility of each source view. ContraNeRF [60] introduces a geometry-aware contrastive loss to improve generalization in the synthetic-to-real setting. However, the methods above do not support rendering a novel view at arbitrary times.

## 2.3 Natural Image Synthesis with Time Translation

Transferring an image to another time zone, for example, from day to night or summer to winter, usually involves style transfer methods. The typical approach is to disentangle content and style information, where style corresponds to the time-specific characteristics. Subsequently, an image is transferred by replacing its original style with the desired time-related style, which is extracted from a reference image [18, 25, 26, 45]. However, this kind of method is limited to domain transfer, lacks the ability to query a specific time, and cannot achieve continuous translation across different time periods. Earlier works [10, 47, 55] on the continuous image translation task assume linear domain manifolds, which may not be suitable for daytime translation. Daytime translation should be cyclic, allowing for translation from day to dusk, dusk to night, night to dawn, and so forth. CoMoGAN [43] proposes the first continuous image translation framework, which enables cyclic or non-linear translation through the design of the functional instance layer and the guidance of a non-neural physical model. Besides, some research efforts are focusing on timelapse generation [1, 4, 5, 37, 59], aiming to generate a timelapse video based on a single source image. However, none of these works are equipped for time transitions in a 3D space, which enhances the overall immersive experience in digital realms, especially in the realm of the metaverse.

## 3 PROPOSED METHOD

Our goal is to synthesize novel views at any given time by learning implicit scene representations across time from sparse input views that are not captured at the desired moment. We begin by reviewing NeRF and highlighting the distinctions in our approach (section 3.1). After detailing our training process (section 3.2), we introduce our proposed framework. As depicted in Fig. 2, our framework comprises five main components: 1) extracting geometry features for each source view (section 3.3), 2) constructing a content radiance field (section 3.4) which stores densities and content features, whose environmental change factors have been excluded, 3) estimating time and extracting time-irrelevant factors (section 3.5), 4) transferring content features at 3D locations to RGB colors of the specific time point and complete a time-dependent radiance field (section 3.6), and 5) rendering a novel view at the desired time through volume rendering (section 3.1). Finally, the proposed loss functions are designed to further improve cyclic changes and enhance the content adaptation based on the input data (section 3.7). Due to space constraints, an in-depth description of the network architecture and extra discussion are relegated to the supplementary

material. The codebase is ready to open and will be made publicly available in due course.

## 3.1 Preliminaries

First of all, we briefly review the idea of NeRF [35]. NeRF optimizes an MLP, whose input consists of 3D spatial location $\mathbf{x}$ and viewing direction $\mathbf{d}$, and whose output corresponds to colors $\mathbf{c}$ and densities $\sigma$, to represent a scene implicitly. In other words, it aims to learn a continuous function: $(\mathbf{c}, \sigma) = F_\theta(\mathbf{x}, \mathbf{d})$. To render a pixel in an image, NeRF first samples $M$ points on the corresponding 3D ray $\mathbf{r}$ and obtains the colors and densities of these sample points. Then, NeRF renders the 2D-pixel color using the volume rendering technique; the formula is described as follows.

$$\hat{C}(\mathbf{r}) = \sum_{i=1}^{M} T_i \left(1 - \exp\left(-\sigma_i \delta_i\right)\right) \mathbf{c}_i. \qquad (1)$$

$$T_i = \exp\left(-\sum_{j=1}^{i-1} \sigma_j \delta_j\right), \qquad (2)$$

where $M$ is the number of sample points along a ray $\mathbf{r}$ and $(\mathbf{c}_i, \sigma_i)$ represent the color and density of point $\mathbf{x}_i$, respectively. $\delta_i$ denotes the distance of the adjacent sample points.

In our method, we introduce novel modifications to the original NeRF. First, following IBRNet [54], $\delta_i$ in eq. (2) is removed for better generalizability. Second, to enable generalizability, we replace the position $\mathbf{x}$ in the model's input with the feature $\mathbf{f^x}$, which is derived from the interpolation of features extracted from input views at the projection pixels corresponding to the 3D position $\mathbf{x}$. Third, we aim to render a novel view at the arbitrary time $t$. Therefore, our TimeNeRF is designed to learn a continuous function: $(\mathbf{c}_t, \sigma) = F_\phi(\mathbf{f^x}, \mathbf{d}, t)$, where $\mathbf{c}_t$ is the color in direction $\mathbf{d}$ at position $\mathbf{x}$ and time $t$ within a scene.

## 3.2 Training Process

We employ a two-stage training process to effectively disentangle content and environmental change factors and subsequently construct implicit scene representations. In Stage 1, we achieve feature disentanglement by leveraging existing style translation models comprising three modules: content extractor, style extractor, and generator. Specifically, we train DRIT++ [26] on the Ithaca365 dataset [9], which contains images across varied weather and nighttime conditions. To enhance the content extractor in DRIT++, instead of only using the content extractor's output, we extract/select content features from 3 different convolutional layers corresponding to different semantic levels for styled image generation. This strategy refines early-layer content extraction, as these features are utilized to generate images, thereby boosting the model's overall content extraction ability.

After Stage 1 training, DRIT++ offers disentangled content features and enables the creation of stylized images as pseudo ground truth for Stage 2 training. In Stage 2, we train our TimeNeRF (detailed in section 3.3 to section 3.6) using input (source) views from the Ithaca365 dataset and using reference images from the Waymo dataset [51]. By merging source images' content features with reference images' style features for training guidance, we can learn

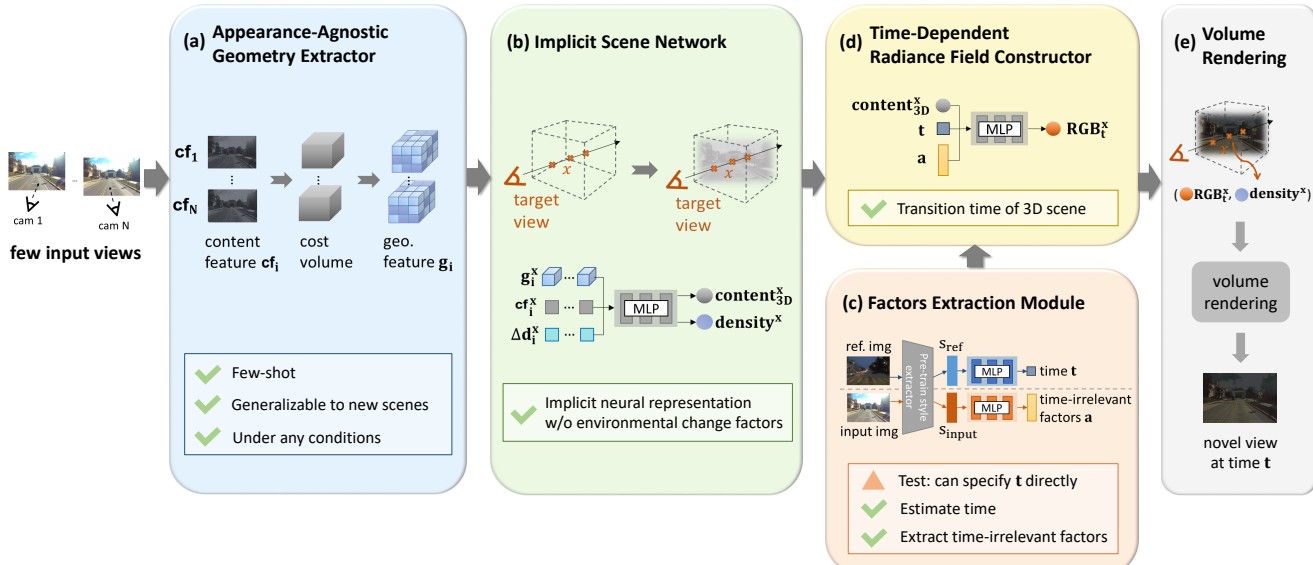

**Figure 2: Architecture Overview.** The proposed framework comprises five main parts: (a) The appearance-agnostic geometry extractor is designed to extract geometry features for each input view. The design of this module allows the model to operate in a few-shot setting without the need for per-scene optimization. It constructs cost volumes and works under various capture conditions by utilizing the content features (section 3.3). (b) For each sample point x, the implicit scene network predicts its corresponding content feature and density by aggregating geometry features, content features, and viewing directions, thus constructing a content radiance field (section 3.4). (c) The factor extraction module predicts both the time and time-irrelevant features. Note that time prediction is only required during training (section 3.5). (d) The time-dependent radiance field constructor transforms the content radiance field into the time-dependent radiance field based on the information from the factor extraction module (section 3.6). (e) Finally, a novel view at time $t$ is rendered via standard volume rendering.

the implicit 3D scene representations across time $F_\phi(\cdot)$. The reason behind using the two datasets is elaborated in section 4.1.

### 3.3 Appearance-Agnostic Geometry Extractor

Given input views $\{I_i\}_{i=1}^N$ and their corresponding camera parameters $\{E_i, K_i\}_{i=1}^N$, we extract content features at three levels $\{cf_i^{(l)}\}_{i=1}^N$ (i.e., $l = 0, 1, 2$ and $l = 0$ is the lowest level feature) for the $N$ input views using the pre-trained content extractor. The content features exclude environmental change factors, such as temporal and weather factors, because the image style has been removed using the pre-trained model. This allows us to handle input views under varying capture conditions. Next, for each input view $v$, we select $S$ nearby views and construct cost volumes [11, 19, 30, 62] at three levels $\{C_v^{(l)}\}_{l=0}^2$ by warping the content features of the $S$ nearby views to align with view $v$ using the corresponding camera parameters. Finally, the cost volumes $\{C_v^{(l)}\}_{l=0}^2$ are put into a 3D-Unet to obtain geometry features $\{g_v^{(l)}\}_{l=0}^2$ for the input view $v$. In the following, we denote $\{cf_i^{(l)}\}_{l=0}^2$ as $cf_i$ and $\{g_i^{(l)}\}_{l=0}^2$ as $g_i$.

### 3.4 Implicit Scene Network

After obtaining geometry features, we may use them to predict colors and densities (section 3.1). However, this would lose the

ability of modeling style change over time. Instead, we propose constructing a content radiance field, an implicit scene representation without environmental change factors. To achieve this, inspired by GeoNeRF [19], we aggregate the extracted geometry features to predict each sample point's density and content features. For a sample point $\mathbf{x} \in \mathbb{R}^3$ along a ray $\mathbf{r}$, we interpolate geometry features $\{g_i\}_{i=1}^N$ and 2D content features $\{cf_i\}_{i=1}^N$, as opposed to the 2D features used in GeoNeRF, to get its corresponding features $\{g_i^\mathbf{x}\}_{i=1}^N$ and $\{cf_i^\mathbf{x}\}_{i=1}^N$ at point $\mathbf{x}$. These are then aggregated to estimate the density and content feature for $\mathbf{x}$. The procedure is described below.

First, we aggregate features via fully-connected layers and multi-head attention layers [53], which facilitate the exchange of information between different views.

$$\sigma^\mathbf{x}, \{\tilde{w}_i^\mathbf{x}\}_{i=1}^N = H(\{cf_i^\mathbf{x}\}_{i=1}^N, \{g_i^\mathbf{x}\}_{i=1}^N), \qquad (3)$$

where $H$ consists of fully-connected layers and multi-head attention layers. $\sigma^\mathbf{x}$ is the density of $\mathbf{x}$ and $\{\tilde{w}_i^\mathbf{x}\}_{i=1}^N$ denotes the enhanced features of $\mathbf{x}$ for each input view. To obtain the 3D content feature $\xi^\mathbf{x}$ for a point $\mathbf{x} \in \mathbb{R}^3$, we predict the weights of input views $\{w_i^{\mathbf{x},v}\}_{i=1}^N$ and use them to calculate the weighted sum of 2D content features $\{cf_i^\mathbf{x}\}_{i=1}^N$. That is

$$\{w_i^{\mathbf{x},v}\}_{i=1}^N = \text{softmax}\left(\text{MLP}_w\left(\{\tilde{w}_i^\mathbf{x} \| \Delta d_i^\mathbf{x}\}_{i=1}^N\right)\right), \qquad (4)$$

 

$$\xi^{\mathbf{x}} = \sum_{i=1}^{N} w_i^{\mathbf{x},v} \cdot cf_i^{\mathbf{x}}, \tag{5}$$

where $\Delta d_i^{\mathbf{x}}$ represents the direction difference between the query view $v$ and input view $i$ by computing the cosine similarity, and the concatenation is denoted by $||$. We predict the weights via an MLP that considers both the direction difference $\Delta d_i^{\mathbf{x}}$ and $\tilde{w}_i^{\mathbf{x}}$ in eq. (4). $\xi$ stands for $\{\xi^{(l)}\}_{l=0}^{2}$. Each individual 3D content feature $\xi^{(l)}$ is computed from the 2D content features $\{cf_i^{(l)}\}_{i=1}^{N}$ at level $l$ by eq. (5).

## 3.5 Factors Extraction Module

To infuse environmental change information into the content radiance field, one intuitive approach is to use the style feature extracted from the pre-trained style extractor. The style feature inherently contains both time-relevant and time-irrelevant environmental factors since the translation model is trained to transfer images to diverse weather conditions and varying times. However, to purely manipulate the timing of radiance fields, it's imperative to separate time-related and time-irrelevant information further from the style feature. To this end, we employ two MLPs: $g_t(\cdot)$ predict time information $t$ and $g_a(\cdot)$ extracts time-irrelevant features $a$ from an image. During training, we extract $t$ from a reference image and $a$ from input images.

To encapsulate the cyclical progression of a day, we design $g_t(\cdot)$ to map the style feature of a reference image onto the interval $[0, 2\pi]$, symbolizing the entire 24-hour cycle. The mapping is trained in an unsupervised manner, utilizing our pseudo stylized image loss $L_{stylemse}$ (eq. (8)) to guide the model in extracting time-related information from reference images and mapping it to the time range $[0, 2\pi]$. By this design, without time labels for training, we can still simulate the variations occurring at different times within a day. After training, the transition of a day is encoded within $[0, 2\pi]$, enabling us to recreate scenes at a desired time $t$ by tuning $t$ during testing.

## 3.6 Time-Dependent Radiance Field Constructor

The objective of time-dependent radiance field constructor "$T$" is to transform the 3D content feature $\xi^{\mathbf{x}}$ into a time-variant color $c_t^{\mathbf{x}}$ for a point $\mathbf{x}$. This involves fusing $\xi^{\mathbf{x}}$ with time-irrelevant environmental features $a$ of input views and time $t$. Essentially, we learn a mapping: $c_t^{\mathbf{x}} = T(\xi^{\mathbf{x}}, a, t)$.

The design of $T$ is based on two main ideas: Firstly, we embed $t$ into $(cos(t), sin(t))$, ensuring uniqueness for each value in $[0, 2\pi]$. This, in conjunction with $L_{\Delta t}$ (eq. (10)), facilitates cyclic changes. Secondly, to ensure the model focuses on learning variations based on time, we introduce a two-branch network. The first branch, fusing $\xi$ with $t$, acts as the template for the change over time. The second branch, merging $\xi$ with both $t$ and $a$, is to further tune the color according to time-irrelevant features $a$.

The network (detailed in Supplementary) is designed in a coarse-to-fine manner. We exploit content features at three different levels $\{\xi^{(l)}\}_{l=0}^{2}$. The predicted color $c_t$ is refined progressively by integrating high-level features down to low-level information. Specifically, we obtain the feature of branch 1, $f_1$, by combining content features

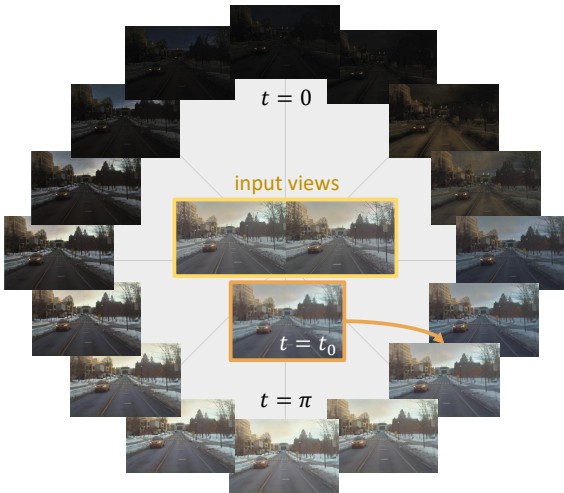

**Figure 3: Novel view synthesis across times. The images in the yellow box represent the two input views of a test scene. The images around the circle are novel views at different times. The image in the orange box is synthesized for the time of input views $t_0$ (eq. (11)), whose image style is consistent with input views.**

from three levels with time $t$. Meanwhile, we derive the feature of branch 2, $f_2$, by integrating content features from three levels with both time $t$ and time-irrelevant features $a$. Finally, we combine $f_1$ and $f_2$ to produce the color $c_t$.

## 3.7 Loss Functions

**MSE loss.** The mean square error loss $\mathcal{L}_{mse}$ in eq. (7) ensures accurate 3D scene construction in our model. Different from the colors $c_t$ described in section 3.6, the predicted colors $c$ in eq. (6) for calculating $\hat{C}(\mathbf{r})$ in this loss are derived from the weighted sum of the original input image colors $\{I_i\}_{i=1}^{N}$, aiding in the learning of densities $\sigma$ (eq. (3)) and weights $\{w_i\}_{i=1}^{N}$ (eq. (4)).

$$\mathbf{c}^{\mathbf{x}} = \sum_{i=1}^{N} w_i^{\mathbf{x},v} \cdot I_i^{\mathbf{x}}. \tag{6}$$

$$\mathcal{L}_{mse} = \frac{1}{|R|} \sum_{\mathbf{r} \in R} \|\hat{C}(\mathbf{r}) - C(\mathbf{r})\|_2^2, \tag{7}$$

where $R$ is the set of rays in each training batch. $I_i^{\mathbf{x}}$ is the projection color of point $\mathbf{x}$ to image $I_i$. $\hat{C}(\mathbf{r})$ is the predicted color computed by eq. (1) using color $\mathbf{c}$ in eq. (6); $C(\mathbf{r})$ is the ground truth color of the target view.

**Pseudo stylized loss.** The pseudo stylized loss $\mathcal{L}_{stylemse}$ aims to guide the model to learn temporal transition by minimizing the difference between our predicted time-dependent pixel color and the pseudo ground truth colors.

$$\mathcal{L}_{stylemse} = \frac{1}{|R|} \sum_{\mathbf{r} \in R} \|\hat{C}_t(\mathbf{r}) - C_{pseudo}(\mathbf{r})\|_2^2, \tag{8}$$

where $\hat{C}_t(\mathbf{r})$ is the predicted color computed by eq. (1) using $c_t$ illustrated in section 3.6, and $C_{pseudo}(\mathbf{r})$ is the pixel color of the

pseudo ground truth, which is the style-transferred output via the pre-trained translation model (DRIT++) given the target view and reference image.

**Loss term of $\Delta t$.** Besides the design where we transform time $t$ into $(\cos(t), \sin(t))$ to ensure cyclic changes, we introduce the $\Delta t$ loss. This loss function in eq. (10) leverages a small MLP, denoted as $D$, to estimate the time difference based on color variations. The predicted time difference is then compared to the pseudo time difference $\Delta t$:

$$\Delta t = \begin{cases} |t - t'| & \text{if } |t - t'| \leq \pi \\ 2\pi - |t - t'| & \text{if } |t - t'| > \pi. \end{cases} \tag{9}$$

$$\mathcal{L}_{\Delta t} = \|D(\mathbf{c}_t, \mathbf{c}_{t'}) - \Delta t\|_2^2, \tag{10}$$

where $t$ and $t'$ are randomly sampled from $[0, 2\pi)$, and $\mathbf{c}_t$ and $\mathbf{c}_{t'}$ are the predicted colors at times $t$ and $t'$, respectively. By minimizing this loss, our color predictions are guaranteed to align with time-based changes.

**Reconstruction loss.** This loss is to ensure that rendered views match the original appearance when the input time aligns with the capture time. This further allows the rendered views of different times to adapt to the inputs. Particularly,

$$t_0 = g_t(s_I), \text{ and} \tag{11}$$

$$\mathcal{L}_{t_0 rec} = \frac{1}{|R|} \sum_{\mathbf{r} \in R} \|\hat{C}_{t_0}(\mathbf{r}) - C(\mathbf{r})\|_2^2, \tag{12}$$

where $t_0$ is derived from predicting the time of an input view $I$ (eq. (11)) and $s_I$ is the style feature of $I$.

**Total loss.** The total loss is as follows:

$$\mathcal{L}_{total} = \mathcal{L}_{mse} + \lambda_1 \mathcal{L}_{stylemse} + \lambda_2 \mathcal{L}_{\Delta t} + \lambda_3 \mathcal{L}_{t_0 rec}, \tag{13}$$

where we set $\lambda_1 = 0.5$, $\lambda_2 = 0.01$, and $\lambda_3 = 0.5$.

# 4 EXPERIMENTS

## 4.1 Datasets

Collecting a dataset with different views and diverse capture times is challenging and unavailable. To address this, we train our model using both the Ithaca365 dataset [9] and the Waymo dataset [51]. Ithaca365 provides different capture views but limited time variations, while Waymo offers images from different times (day, dusk, dawn, and night) on sunny days. From Ithaca365, which includes conditions like sunny, cloudy, rainy, snowy, and nighttime, we randomly selected 2180 scenes with 3 distinct views. This allows TimeNeRF to be versatile for various conditions during testing. On the other hand, reference images for learning time information are taken from Waymo. In all experiments, TimeNeRF is trained with the aforementioned dataset configurations. For testing, we evaluate the model's generalizability using scenes outside the training set, as well as the T&T dataset [22] and the LLFF dataset [34].

## 4.2 Implementation Details

During training, we use $N = 2$ input views due to the Ithaca365 dataset's limitations of having only 4 distinct viewpoints per scene, yet 2 of them are nearly overlapping. However, our model is flexible enough to handle more views during testing. We train TimeNeRF over 15 epochs, sampling 1024 rays as the training batch. The number of sample points $M$ is set to 128. The Adam optimizer [21] is applied with a $5 \times 10^{-4}$ learning rate and a cosine scheduler without restarting the optimizer [32]. In the testing phase, we neither re-train nor fine-tune our model for new scenes.

## 4.3 Novel View Synthesis Across Time

To show TimeNeRF's capability of rendering novel views over varying times, we use the same input views to render 16 novel views, querying the same target viewpoint but specifying times as $t = \left\{ \frac{i}{16} \cdot 2\pi \right\}_{i=0}^{15}$. As shown in Fig. 3, TimeNeRF correctly produces distinct appearances for each specific time, capturing transitions that resemble the natural shifts seen throughout the day.

**Comparison.** To the best of our knowledge, no existing method is designed for rendering novel views across time. For comparison, we combine view synthesis algorithms with image translation models, which transform images to reflect different times of the day. We utilize our model to produce novel views. Next, DRIT++ [26], HiDT [1], and CoMoGAN [43] are used to transfer images across times. Note that **DRIT++ and HiDT require extra reference images for the desired time-relevance features**. Thus, we use frames from 24-hour time-lapse videos as reference images to generate images spanning an entire day. **Conversely, CoMoGAN, similar to our approach, can directly specify times**. The results in Fig. 4 synthesize the *playground* scene in the T&T dataset using 3 input views. While these methods can generate variations based on time, they sometimes yield undesirable effects, such as incorrect black patches during nighttime and color bias on the ground. Furthermore, all of these methods have the cross-view appearance inconsistency issue (geometric inconsistency), detailed in section 4.4.

In comparison to previously discussed methods, our approach achieves more accurate cyclic appearance changes through consistent time translation. To validate this, we conducted a comprehensive user study. In this study, we presented participants with transformation results at eight distinct times of day: pre-dawn, dawn, mid-morning, afternoon, dusk, evening, and late night. These results, generated by our method and other referenced methods, were shown alongside a reference image corresponding to each specific time. Participants were tasked with identifying the method that ensured consistent translation between frames and most accurately matched the time depicted in each reference image. The findings of this study, depicted in Fig. 7, demonstrate that our method facilitates smoother transitions across different times of the day. Please refer to the supplementary materials for more analysis.

**Generalizability.** Additionally, we evaluate our model on the T&T dataset [22] in Fig. 4, demonstrating its adaptability to different datasets. More results under varying conditions are shown in the supplementary material.

## 4.4 View Consistency

Following [6, 15, 16], we employ the warped LPIPS metrics [65] and RMSE to measure consistency across different views. The score is computed by $E(O_v, O_{v'}) = f(O_v, W(O_{v'}), M_{v,v'})$, where $O_v$ and $O_{v'}$ represents the generated image $\hat{I}_v$, $\hat{I}_{v'}$, and their camera parameters. We warp the result from view $v'$ to view $v$ using the depth estimation in our model, which is estimated by replacing $\mathbf{c}_i$ in eq. (1) to the

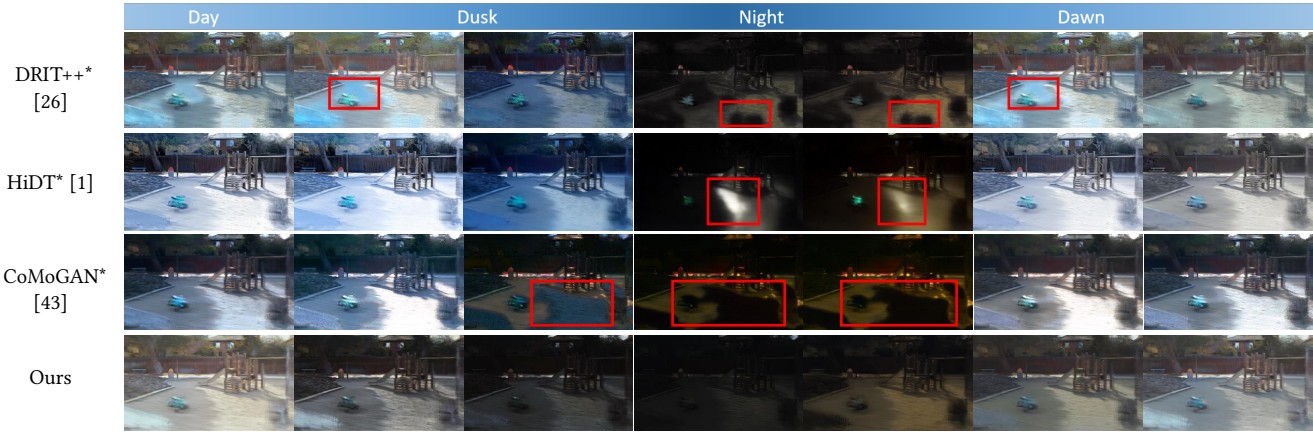

Figure 4: Comparison of view synthesis across time. We generate novel views at 7 different times to show the cyclic changes of a day from 3 input views. Unlike our approach, other methods first need to utilize the view synthesis model to render the novel view before executing time transitions, denoted by *. Besides, these methods may produce color bias and incorrect dark areas on the ground.

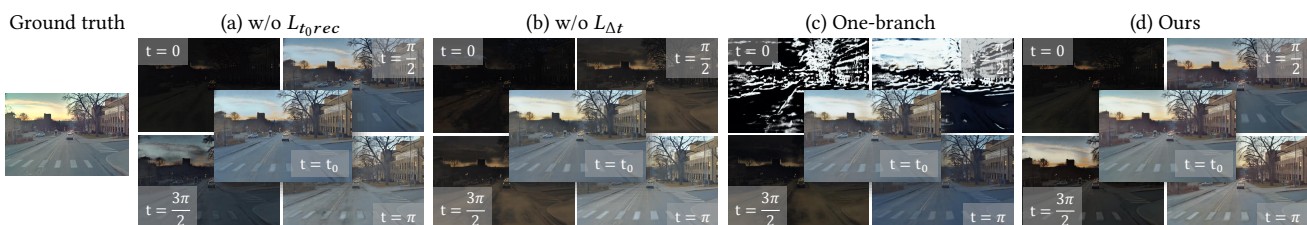

Figure 5: Ablation study. (a) shows the result when $L_{t_0 rec}$ is omitted (eq. (12)) (b) illustrates the outcome without $L_{\Delta t}$ (eq. (10)). (c) depicts the result when the two-branch network is not utilized (section 3.6). Finally, (d) is the result from our complete framework.

| Method | Ithaca365 [9] | | T&T [22] | | | | | | | | | | | |
| | Average | | Family | | Horse | | Playground | | Train | | Average | | | |
| | LPIPS↓ | RMSE↓ | LPIPS↓ | RMSE↓ | LPIPS↓ | RMSE↓ | LPIPS↓ | RMSE↓ | LPIPS↓ | RMSE↓ | LPIPS↓ | RMSE↓ | | |
| DRIT++*[26] | 0.098 | 0.061 | 0.094 | 0.076 | 0.056 | 0.110 | 0.084 | 0.064 | 0.052 | 0.096 | 0.072 | 0.086 | | |
| HiDT*[1] | 0.082 | 0.065 | 0.102 | 0.090 | 0.069 | 0.126 | 0.076 | 0.070 | 0.057 | 0.115 | 0.076 | 0.100 | | |
| CoMoGAN*[43] | 0.164 | 0.070 | 0.138 | 0.086 | 0.088 | 0.125 | 0.144 | 0.074 | 0.087 | 0.088 | 0.114 | 0.093 | | |
| Ours | **0.058** | **0.038** | **0.066** | **0.045** | **0.034** | **0.064** | **0.053** | **0.040** | **0.036** | **0.058** | **0.047** | **0.052** | | |

Table 1: Comparison of view consistency. We evaluate the consistency scores (LPIPS and RMSE) using 15 scenes from the Ithaca365 dataset, each with 2 target views; 4 scenes from the T&T dataset, each with 15 pairs of target views. For each target view, we generate results at 16 different time points. Our method achieves better cross-view consistency in different datasets.

depths of sample points. This process is denoted by $W(O_{v'})$. $M_{v,v'}$ is a mask of valid pixels and $f$ is the measurement metric, such as LPIPS or RMSE. Both quantitative and qualitative analyses (table 1 and Fig. 6) demonstrate that our method achieves better cross-view consistency. This improvement stems from our innovative approach of directly modeling time-relevant appearance changes within a 3D space, in contrast to the image translation approaches that operate in 2D space.

## 4.5 Few-Shot View Synthesis

To demonstrate TimeNeRF's capability in the traditional novel view synthesis task, we evaluate it on the Ithaca365 and LLFF datasets [34], comparing its performance with other few-shot generalizable 3D modeling approaches. For a fair comparison, we train all the models on the Ithaca365 dataset. The quantitative comparison for the Ithaca365 and LLFF datasets is presented in table 2. While TimeNeRF's primary innovation lies in modeling the temporal dynamics of 3D scenes, this experiment reveal that our model

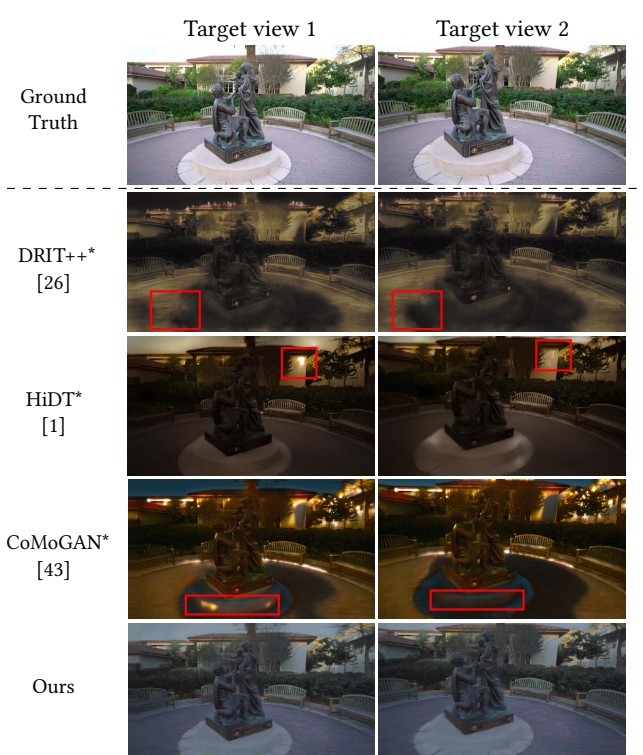

Target view 1 Target view 2

Ground Truth

DRIT++*
[26]

HiDT*
[1]

CoMoGAN*
[43]

Ours

**Figure 6: View inconsistency issue. We render 2 target views corresponding to the same time using the same set of 3 input views. The red boxes highlight the inconsistent regions between views. Among these results, our method produces more consistent views.**

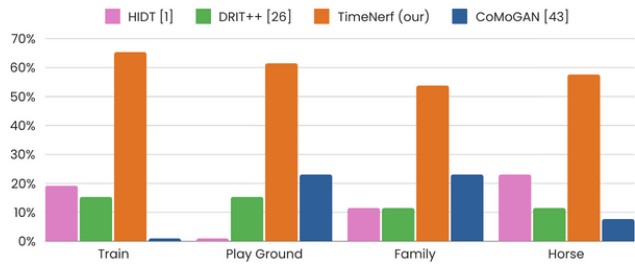

**Figure 7: Novel view synthesis across times. We conduct a user study to ask subjects to select results that are more consistent translation between frames and accurately corresponds to the time depicted in the reference image on T&T dataset. The number represents the percentage of preference based on 25 trial participants.**

outperform MVSNeRF and matches the capabilities of GeoNeRF in in view synthesis.

| Method | Ithaca365 [9] | | | LLFF [34] | | |
|---|---|---|---|---|---|---|
| | PSNR↑ | SSIM↑ | LPIPS↓ | PSNR↑ | SSIM↑ | LPIPS↓ |
| MVSNeRF[2] | 25.90 | 0.703 | 0.485 | 16.58 | 0.513 | 0.503 |
| GeoNeRF[19] | 27.46 | 0.768 | **0.364** | 19.62 | **0.597** | **0.415** |
| Ours | **27.67** | **0.773** | 0.367 | **19.71** | 0.593 | 0.418 |

**Table 2: Novel view synthesis. The best and the second best result are highlighted in bold and underline, respectively.**

## 4.6 Ablation Study

**The designed loss functions.** We study the effectiveness of our designed loss functions. The reconstruction loss is designed to ensure that the model renders views with accurate appearances when the input time coincides with the input views' capture time. Moreover, it allows the rendered views to adapt based on these inputs for different times. In Fig. 5(a), the model without $L_{t_0}$ generates a road that appears blue at $t = t_0$, although it should be gray. The delta t loss aims to ensure that the model produces unique outputs for each $t \in [0, 2\pi]$. Fig. 5(b) shows the results obtained when training the model without $L_{\Delta t}$. We observe that the model renders similar outputs at $t = \frac{\pi}{2}$ and $t = \frac{3\pi}{2}$.

**Two-branch network.** In Fig. 5(c), we note that omitting the two-branch network from our time-dependent radiance field constructor (section 3.6) leads to mapping failures within $[0, 2\pi]$. The model generates nearly black-and-white scenes at $t = 0$ and $t = \frac{\pi}{2}$. This limitation is due to the lack of the first branch, which serves as a template to constrain the output. By incorporating our proposed two-branch network and the designed loss functions (Fig. 5(d)), we achieve consistent and accurate results across all time points.

## 5 CONCLUSION AND DISCUSSION

We propose TimeNeRF, a novel framework for view synthesis that renders views at any time from limited input views without per-scene optimization. We have effectively leveraged existing datasets to address the lack of data availability for the aforementioned task. Our model transitions smoothly across time by creating a content radiance field and transforming it into a time-dependent radiance field. In addition, our designed loss functions ensure cyclic changes and adaptive results based on inputs. Evaluations show TimeNeRF's capability to produce photorealistic views across time.

As the initial solution in few-shot, generalizable novel view synthesis across arbitrary viewpoints and times, TimeNeRF has showcased its efficacy on various datasets. Nevertheless, significant opportunities for advancement remain. For a comprehensive 3D scene model, the dynamics of lighting angles from diverse sources—such as the sun and streetlights—and the associated constraints of object shadows relative to these light sources need to be accounted for. Additionally, achieving appearance and view consistency in both static and dynamic objects throughout varying times of the day poses a substantial challenge that could dramatically enhance user experience. Moreover, incorporating diffusion models as a tool for data augmentation presents a promising direction to address the scarcity of available data, which could catalyze further enhancements in model performance. These considerations may underline future research on 3D modeling technologies.

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
