# OpenReview forum: "TimeNeRF: Building Generalizable Neural Radiance Fields across Time from Few-Shot Input Views"
_acmmm.org/ACMMM/2024/Conference — MM2024 Oral_

### Official Review · Reviewer_uQTU · 2024-05-21

**Rating:** 4
**Confidence:** 1

**Summary:**

This paper proposes TimeNeRF for rendering novel views at any viewing angle and at any time.

**Strengths:**

+ A new setting.
+ A novel NeRF-based approach that renders novel views at arbitrary times from few inputs and achieves generalizability to new scenes captured under any conditions.

**Limitations:**

+ The method section write is very poor, many places lack some explanation, I give some examples here:
   + The DRIT (see line 332 on page 3 of the paper) is cited in many places, but no explanation is provided in the paper.
   + There are no details on how to build the cost.
+ Lack of quantitative ablation results, it is difficult to determine whether the proposed module is effective, especially in Figure 5 (a) and Figure 5 (d).
+ Does this task have corresponding ground truth (as shown in Figure 4)? If it is a style transfer task, changing the rendering color of the scene at a certain moment, I strongly recommend conducting user studies. This is because such tasks are highly subjective, and it is difficult to determine which result is better based solely on visual results.
+ Although the proposed method can effectively transform colors based on time, it does not provide specific time points. What criteria are used to determine these time points? How does this time-based transformation differ from directly using text-based transformation (such as Instruct-NeRF2NeRF)?

**Suitability:**

2

---

### Official Review · Reviewer_Lem6 · 2024-05-23

**Rating:** 4
**Confidence:** 3

**Summary:**

TimeNeRF can generate novel views of scenes at any time of day with minimal input views, without requiring per-scene optimization. It combines multi-view stereo, nerf, and disentanglement strategies to create an implicit radiance field that generalizes in a few-shot setting.  TimeNeRF excels in rendering realistic transitions between day and night, capturing detailed scene changes smoothly, compated to GAN based methods.

**Strengths:**

1. The topic addresses a good setting by focusing on view synthesis that not only generates new perspectives but also smoothly transitions across different times of the day. This adds a temporal dimension to 3D scene modeling, enhancing its applicability in dynamic real-world scenarios and virtual environments like the metaverse.

2. The method achieves generalizability by leveraging multi-view stereo, neural radiance fields, and disentanglement strategies. This allows it to adapt to new scenes captured under various conditions with minimal input views, making it robust and versatile for diverse applications.

3. The model's optimization process does not depend on precise time-stamped data, enabling it to render realistic scene changes without the need for exact temporal alignment. This flexibility simplifies data collection and broadens the model's usability across different temporal contexts and environments.

**Limitations:**

1. It is a weakness that the method did not include comparisons with other NeRF-based approaches. Instead, it was compared to 2D generation models, which do not incorporate 3D modeling. This limits the evaluation's relevance, as it does not directly benchmark against state-of-the-art techniques specifically designed for 3D view synthesis.

2. Some straightforward baselines could have been considered, such as integrating instruct-nerf2nerf and few-shot generalizable NeRF methods with style prompts (e.g., "make it dark"). This would provide a more comprehensive assessment by including methods that address similar challenges in temporal scene representation and view synthesis.

3. The claim of time modeling is somewhat misleading. The method seems to focus more on style-conditioned modeling, akin to what is seen in WildNeRF, rather than genuine temporal dynamics. This distinction is important, as true time modeling would require capturing and predicting changes over continuous time rather than applying style variations to static scenes.

**Suitability:**

2

---

### Official Review · Reviewer_Kih4 · 2024-05-24

**Rating:** 5
**Confidence:** 4

**Summary:**

This work introduces TimeNeRF, which can render novel views at different times of the day in a feed-forward manner. Specifically, TimeNeRF starts by disentangling the content features from the input images, which are then used to learn a generalizable NeRF model. To inject the time-dependent information into NeRF, TimeNeRF leverages a factor extraction module to pass the time and related time-irrelevant information before predicting the color for each point. Experiments on several existing benchmarks demonstrate the soundness of the introduced TimeNeRF.

**Strengths:**

* The setting of rendering generalizable NeRF at different times is interesting.
* It is technically sound to map the style features onto a 24-hour cycle, removing the requirement of exact time labels and making the framework more applicable to different datasets.
* The paper is well-written, and it is easy to follow.

**Limitations:**

* The visual results are not surprising enough. I would imagine that for those street views (‘itheca_street.mp4’ in the supp.), the streetlights should be on at night, leaving lights and shadows on the street. Or the shadows cast by the sun should be gone at night (left part of the ‘T&T_train.mp4’). However, there is no such effect in TimeNeRF. The current visual results are very similar to applying a transparent black mask on top of rendered images, making it hard to understand whether TimeNeRF has effectively learned the time-relevant information. Is there any scene that presents contents where the streetlights will be turned on at night?

* It is unclear what is in the $\textrm{content}^x_\textrm{3D}$. Typically, in a 2D image, the content might refer to geometry/structure, and style refers to color information. However, in the NeRF representation, geometry is already captured by the density, so I am curious about what is being captured in the $\textrm{content}^x_\textrm{3D}$ intuitively. Besides, I can see that ray direction is still fed into the Implicit Scene Network, does it mean that the $\textrm{content}^x_\textrm{3D}$ is also view-dependent, similar to the color of the original NeRF? The authors are recommended to volume render the $\textrm{content}^x_\textrm{3D}$ at different viewpoints and further analyse what kinds of physical attributes $\textrm{content}^x_\textrm{3D}$ aims to learn.

**Suitability:**

3

---

### Official Review · Reviewer_FwYq · 2024-05-26

**Rating:** 3
**Confidence:** 4

**Summary:**

This paper proposes a method to solve time-related appearance changes using few-views for novel-view synthesis. More specifically, they solve a challenging problem: "How to render novel-views at different time"?  Previous approaches like NeRF-W and Ha-NeRF represent the underlying geometry under different illumination conditions and appearance changes. However, they depend on appearance codes and per-scene optimization. The proposed method can render novel views simultaneously without relying on reference images or appearance codes.

**Strengths:**

- The proposed method requires only a few views to learn geometry and appearance changes.
- Two-stage training process to disentangle content and environmental factors.
- The proposed method outperforms baselines, which combine view-synthesis and image translation methods.

**Limitations:**

- **Blurry and inconsistent output in the supplementary videos**: In the videos shared in the supplementary, there are clear artifacts in the rendered novel views. For example, in "T&T_playground.mp4", the structure in the playground is blurry and has visual artifacts.

- **Missing comparison with Instruct NeRF2NeRF(IN2N)**: The proposed work proposes a method to generate novel-views with time-varying appearance changes. It's also important to compare with works like IN2N, which are capable of editing the appearance of a scene.

- **How about effects like rain/fog/snow?:** Although this is not a limitation, can the proposed work also handle these changes? This will further show that the proposed method is generalizable to different scenarios.

- **Missing details like training time**: Details like training time, no. of parameters and inference time are not discussed in the current manuscript. Authors should discuss this.

- **Missing details for 3D-Unet mentioned in L393-397**: Proposed method does not discuss the 3D-Unet details which is used to obtain geometric features.

**Suitability:**

3

---

### Meta-Review · Area_Chair_NVnA · 2024-06-27

**Recommendation:** Accept (Oral)
**Confidence:** 5

**Metareview:**

This paper targets at the problem of learning dynamic NeRF with few-shot views, which is quite interesting. All the reviewers agree to accept the manuscript. I recommend a decision of acceptance. The authors should include the comparisons and some details in the final version.